# A Network Pharmacology Identified Metastasis Target for Oral Squamous Cell Carcinoma Originating from Breast Cancer with a Potential Inhibitor from *F. sargassaceae*

**DOI:** 10.3390/ph17101309

**Published:** 2024-09-30

**Authors:** Abdullah Alqarni, Jagadish Hosmani, Saeed Alassiri, Ali Mosfer A. Alqahtani, Hassan Ahmed Assiri

**Affiliations:** Department of Diagnostic Dental Sciences & Oral Biology, College of Dentistry, King Khalid University, Abha 61421, Saudi Arabia; aawan@kku.edu.sa (A.A.); sadelboh@kku.edu.sa (S.A.); al.alqahtani@kku.edu.sa (A.M.A.A.); halmuawad@kku.edu.sa (H.A.A.)

**Keywords:** metastatic breast cancer, molecular docking, oral squamous cell carcinoma, sargassum

## Abstract

This study aimed to identify specific therapeutic targets for oral squamous cell carcinoma (OSCC) that metastasize from breast cancer (BC) by using network pharmacology. The Gene Expression Omnibus for OSCC and BC served as the source of gene expression datasets and their analysis. Upregulated genes and the common intersecting genes of these cancers were determined along with that of the phytochemicals of *F. sargassum* to predict the pharmacological targets. Further, gene enrichment analysis revealed that their metastasis signature and metastasis targets were determined via a protein interaction network. Molecular docking and pharmacokinetic screening determined the potential therapeutic phytochemicals against the targets. The interaction network of 39 genes thus identified encoding proteins revealed HIF1A as a prominent metastasis target due to its high degree of connectivity and its involvement in cancer-related pathways. Molecular docking showed a strong binding affinity of isonahocol D2, a sargassum-derived compound with HIF1A, presenting a binding energy of −7.1 kcal/mol. Further, pharmacokinetic screening showed favorable ADME properties and molecular dynamics simulations showed stable interactions between isonahocol D2 and HIF1A, with significant stability over 100 ns. This study’s results emphasized that isonahocol D2 is a promising therapeutic candidate against HIF1A in OSCC metastasized from breast cancer in translational medicine.

## 1. Introduction

Oral squamous cell carcinoma (OSCC), which metastasizes from breast cancer (BC), presents a complex challenge in oncology due to its relatively rare occurrence. Metastasis in the oral cavity is uncommon, comprising only 1–1.5% of all tumors [1]. Notably, in women, cancer of the breast is the primary leading cause of oral metastases, with around 41% affecting the jawbone and 24.3% affecting soft tissues [2]. Globally, breast cancer is responsible for 1.9 million cases and 601,000 deaths. It is one of the most frequently diagnosed cancers in the United States and the second most significant cause of cancer-related deaths [3]. On the other hand, OSCC represents the most important form of oral cancer, comprising around 90% of oral malignancies. OSCC originates from squamous epithelial cells that affect various parts of the oral cavity, like the mouth, tongue, lips, gingiva, and palate [4]. Oral metastases from breast cancer are complex molecular processes, highlighting the importance of understanding and addressing this clinical entity. Oral metastases account for approximately 1% of all oral malignancies, particularly in the mandible and gingiva [5]. Breast cancer frequently spreads to nearby lymph nodes; it can also metastasize to distant organs, including the bones, liver, lungs, and brain. Although metastasis to the oral cavity is relatively rare [6], given the increase in the widespread frequency of breast cancer with its increasing chance of OSCC metastasis, it cannot be underestimated. 

The landscape of metastatic cancer research is rapidly evolving, driven by significant advancements in understanding the mechanisms of metastasis and developing innovative therapeutic strategies. Recent studies have elucidated the complex processes that enable cancer cells to disseminate from the primary tumor to distant organs, highlighting the critical roles of the tumor microenvironment, genetic and epigenetic alterations, and cellular interactions [7,8]. Metastasis is a major cause of mortality in cancers such as breast, lung, colon, rectum, and prostate [9]. It remains one of the most significant challenges in oncology, with oral squamous cell carcinoma (OSCC) and breast cancer being particularly notorious for their aggressive nature and poor prognosis [10]. 

Despite advancements in early detection and localized treatment, the management of metastatic disease continues to be a significant hurdle. This is especially true for OSCC and breast cancer, where metastasis often leads to a dramatic decline in patient survival rates [11,12]. Murgia et al. provided literature findings concerning oral metastases originating from BC, highlighting clinical and radiographic characteristics, as well as considerations for differential diagnosis [2]. BC diagnosis typically involves imaging studies such as mammography, ultrasound, and MRI, accomplished by tissue biopsy for histopathological analysis [13]. Mammography is the primary screening tool, often followed by ultrasound for further evaluation [14]. A biopsy confirms the diagnosis and provides essential tumor characteristics like hormone receptor status and HER2 expression [14]. OSCC is diagnosed through a clinical examination and biopsy of suspended lesions. Histopathological examination of biopsy specimens confirms the diagnosis and provides critical tumor characteristics [15].

Numerous studies have been carried out significantly for novel therapeutic approaches to enhance the prognosis of cancer patients [16,17,18,19]. Recent studies have revealed the involvement of crucial genes and proteins in breast cancer development [16,17,18,19]. Nevertheless, the mechanism involved in metastasis to distant sites remains poorly understood. Therefore, we are exploring therapeutic targets along with prospective medications for patients with OSCC metastasized from BC. Notably, recent advances in omics technology have facilitated the identification of novel molecular markers for diagnosis and therapeutic targets for diseases of a complex nature, such as cancer, neurological diseases, and heart diseases. Indiraleka et al. [16] identified the druggable natural phytochemical by methods of computation against the GPR116 therapeutic target for Triple Negative Breast Cancer (TNBC). Magesh et al. [17] found 3-O-methyl-d-glucose as the drug candidate to inhibit the oncogenic therapeutic targets EGFR and MAPK in oral cancer. Likewise, Sudhan et al. used a computational protein network and identified a putative biomarker for coronary artery disease [18]. Similarly, Baul et al. showed the molecular interaction of quercetin against the α-synuclein with the potential drug target of Parkinson’s disease [19].

Presently, the primary treatment methods for cancer include radiotherapy, chemotherapy, and standard surgical procedures [20]. Despite the advancement of numerous pharmaceutical interventions for cancer treatment, their effectiveness is often limited due to the significant challenge of drug resistance, reducing their utility as effective therapeutic options [21]. Natural compounds from various sources, including marine environments, have long been recognized for their potential in cancer therapy [22]. Among these, marine-derived compounds have significant attention due to their rich biodiversity and diverse biological activities. Brown algae, in particular, have emerged as promising candidates for cancer treatment, exhibiting anti-cancer, anti-inflammatory, and anti-proliferative properties [23]. Specifically, the anti-cancer effects of sargassum extracts against breast cancer cells attributed to its bioactive compounds, such as fucoxanthin and fucoidan [24], were revealed in studies. According to Murad et al. [25], Sargassum extracts possess a tendency to promote apoptosis and arrest the cell cycle in the cells of breast cancer, thus indicating their benefits as a natural anti-cancer agent.

Henceforth, this study aimed to implement a computational approach integrating the high-throughput gene expression profile, protein interaction network, molecular docking with the sargassum sp. (brown algae) phytochemicals, and molecular dynamics (MD) simulations for the complex protein–ligand molecules. This study established a crucial target protein involved in the OSCC metastasis of breast cancer. In particular, screening the phytochemicals against the crucial therapeutic target protein may benefit patients who have a tendency to develop OSCC metastasized from BC. Also, inhibiting the target protein involved in the metastatic progression of BC to OSCC may potentially improve patient outcomes and enhance overall treatment efficacy.

## 2. Results

### 2.1. Gene Expression Profiling

Four gene expression profiles of OSCC (GSE30784, GSE23558, GSE3524, and GSE10121) were selected and analyzed. The GSE30784 dataset contained 229 samples with 184 OSCC cases and 45 controls, which showed 6583 upregulated genes. Likewise, the GSE23558 dataset contained 32 samples in total, which had 27 OSCC cases, and four control groups showed 3335 upregulated genes. Similarly, the GSE3524 dataset, which comprised 20 samples, with 16 OSCC and four controls, showed 1600 upregulated genes. Also, the dataset GSE10121, consisting of 41 samples, with 35 OSCC and six control sample groups, showed 4617 upregulated genes. Likewise, four transcriptome profiles of breast cancer were selected and analyzed. The GSE10810 contains 58 samples, with 31 tumors and 27 controls, which include 3920 upregulated genes. Similarly, the GSE15852, comprised of 86 samples, with 43 tumors and 43 controls on DEG analysis, showed 1204 upregulated genes. The GSE20086, comprised of 12 samples, with six tumors and six controls, provides 1557 upregulated genes. Also, the GSE42568, with 121 samples, with 104 tumors and 17 controls, showed 5830 upregulated genes. Then, we intersected all upregulated genes in OSCC and BC, providing 5172 common genes between these conditions.

### 2.2. Identification of Drug Targets

The protein targets for the 147 phytochemicals sourced from sargassum (brown algae) were retrieved from a variety of databases, as mentioned in the methodology section. Collectively, 97 protein targets for 147 phytochemicals were obtained. Subsequently, an intersection analysis was conducted between the 97 targets and the shared 5172 upregulated genes of OSSC and BC. On the intersection, 39 targets (APH1A, BCL2L1, CA12, CTSD, FAAH, FDPS, HIF1A, HMGCR, HPRT1, HSD17B10, HSP90AA1, JAK3, KMT5A, MAPK1, MMP3, MSR1, PNP, PTGS1, PTPN1, SQLE, SYK, TDP1, TYMP, TYMS, VDR, VEGFA, CDC25C, KIF20B, MMP13, PDE4D, ALOX5AP, CDK5R1, MGAM, PTAFR, BLM, HPSE, IGF2R, INSR, and JAK1) were identified, which indicates that these common genes of OSCC and BC could respond to any of the 147 phytochemical candidates.

### 2.3. Enrichment Assessment

Gene enrichment analysis was performed on the shared 39 genes, elucidating their roles in various cellular/molecular processes. The outcomes revealed significant associations with the regulation of cell proliferation, phosphorylation, and programmed cell death regulation, as depicted in Figure 1A. In terms of molecular functions, the predominant observations were related to the actions of protein tyrosine kinase and transmembrane receptor protein tyrosine kinase, as illustrated in Figure 1B. Additionally, the analysis of cellular components identified them as integral components within the plasma membrane, as shown in Figure 1C. Molecular pathway analysis showed the involvement of these proteins in pathways such as the PI3K-AKT signaling pathway, HIF-1 signaling pathway, and metabolic pathways, as depicted in Figure 1D. These findings revealed that these 39 proteins are crucial in the cancer-associated process. These findings revealed that these 39 proteins are crucial in the cancer-associated process. Additionally, the metastasis signature of these 39 genes was verified using the CMGene database. Of 39 genes, 21 (BCL2L1, CA12, CTSD, HIF1A, HPSE, HSP90AA1, IGF2R, JAK1, MAPK1, MMP13, MMP3, MSR1, PTAFR, PTGS1, PTPN1, SQLE, SYK, TYMP, TYMS, VDR, and VEGFA) were showed the involvement in the metastasis, whereas the other genes could be the co-regulators of the cancer metastasis as demonstrated through enrichment analysis.

### 2.4. Protein–Protein Interaction Network and Target Screening

Next, the protein–protein interaction (PPI) network was built by importing the 39 targets encoding proteins into the STRING within Cytoscape 3.9.1. The network is constituted of 39 nodes and 63 edges, as illustrated in Figure 2. To identify key proteins within this network, the cytoHubba plug-in was employed, utilizing the method of the node connect degree to rank proteins based on their direct interaction. Consequently, HIF1A emerged as a prominent protein target, exhibiting a high degree of connectivity, with eleven crucial metastatic proteins in the network. Such assessments designate HIF1A as a potential therapeutic target that addresses BC connectivity with the OSCC process. Although these 39 targets could respond to any of the 147 phytochemical candidates, it is vital to find the best phytochemical against HIF1A through molecular docking analysis.

### 2.5. Molecular Docking Assessment

A molecular docking study was conducted to investigate the interactions between HIF1A and 147 phytochemicals derived from sargassum using Maestro 11.2 Schrödinger software. All collected phytochemicals in SDF format were optimized through the Ligprep module to ensure appropriate molecular conformations. Subsequently, a receptor grid was generated around binding sites within the HIF1A protein, which enabled docking at the preferred binding site. Glide docking was performed to assess the binding affinities and poses of the phytochemicals within the receptor grid. The binding efficiencies of phytochemicals against the HIF1A protein around the active site (TYR145, GLN147, PHE100, LEU101, TYR102, GLN239, ARG238, ASP237, PRO235, ASP104, LYS106, LYS107, ARG320, LYS324, GLN203, GLU202, ASP201, HIS199, TRP296, PRO197, THR196, LEU188, SER184, LEU186) were taken. Notably, among the docked phytochemicals, isonahocol D2 exhibited promising interactions with HIF1A with a binding energy of −7.1 kcal/mol, indicating their potential as therapeutic candidates (Figure 3). ADME (Absorption, Distribution, Metabolism, and Excretion) screening was performed using QikProp to assess the pharmacokinetic properties of the isonahocol D2. Interestingly, no violations were found, highlighting its favorable characteristics in terms of pharmacokinetics.

### 2.6. Molecular Dynamics Simulation

The molecular dynamics (MD) simulations with Desmond were used to explore the dynamic behavior of the HIF1A–isonahocol D2 complex. In MD simulations, the time-dependent alteration of the HIF1A–isonahocol D2 complex was computed over 100 ns. We monitored the stability of the complex using root-mean-square deviation (RMSD) analysis, root-mean-square fluctuation (RMSF), and protein–ligand contacts. In RMSD analysis, the HIF1A showed a fluctuation between 35 and 45 ns at 3.6 Å throughout the simulation time. Notably, after 50 ns, the RMSD attained stable conformation (Figure 4A). Similarly, the RMSF analysis of the HIF1A–isonahocol D2 complex showed residue fluctuation during simulation. High fluctuation was noticed at the N-terminal (5.4 Å) as compared to the C-terminal (4.2 Å). The HIF1A amino acid residues at the active sites showed significant levels of fluctuation that enabled effective interaction with isonahocol D2 (Figure 4B). In addition, the ligand RMSF showed the few atoms within the isonahocol D2 that contributed widely to interacting with HIF1A (Figure 4C). Furthermore, the protein–ligand contact map showed amino acids TYR102, ARG238, GLN239, GLU202, and ASP104 were prominently involved in the interaction with isonahocol D2 (Figure 4D). Overall, the MD analysis suggests that isonahocol D2 has significant potential for binding with HIF1A, which leads to the formation of stable complexes throughout the simulation period.

## 3. Discussion

Oral squamous cell carcinoma and breast cancer are two prevalent malignancies with distinct primary sites, but they share some common genetic alterations that may provide metastatic signatures [26]. Both types of cancer demonstrate lymphatic dissemination, often involving regional lymph nodes as a frequent site of metastasis [27,28]. Additionally, breast cancer cells can metastasize to distant organs like the lungs, liver, bones, and brain, leading to advanced-stage disease and a poor prognosis [29]. Despite advances in diagnosis and treatment modalities, metastatic OSCC and BC remain challenging to manage, emphasizing the need for innovative therapeutic approaches that target metastatic progression pathways. The metastasis of OSSC in BC represents unique and clinically significant findings. While OSCC originating in the oral cavity is relatively common, metastasis from distant primary tumors, such as BC, is rare but carries important implications for patient management. Metastatic spread to the oral cavity often indicates advanced disease progression and may present with symptoms such as oral lesions, pain, and difficulty swallowing or speaking [30].

Natural phytochemicals derived from plants have gained significance for their potential therapeutic properties in cancer management, including metastatic conditions [31]. These bioactive phytochemicals, present abundantly in fruits, vegetables, herbs, and spices, exhibit antioxidant, anti-inflammatory, anti-proliferative, and anti-metastatic pharmacological effects [32]. Studies have shown that certain phytochemicals can modulate key signaling pathways involved in cancer metastasis, including those regulating cell adhesion, migration, invasion, and angiogenesis [33]. Moreover, phytochemicals were capable of enhancing the efficacy of conventional cancer therapies and mitigating their adverse effects, highlighting their potential as adjunctive agents in cancer treatment [34]. Traditionally, the approach to cancer care involves detecting tumor lesions using suitable diagnostic imaging techniques and then employing treatments such as chemotherapy, radiotherapy, or surgery [35]. Nevertheless, these treatments come with drawbacks like incomplete tumor removal during surgery, unintended toxic effects, insufficient drug concentration at the site of disease, and challenges in drug delivery to tumors due to abnormal blood vessel structure, leading to increased interstitial pressure and reduced blood flow [36]. Interestingly, marine phytochemicals derived from seaweeds, algae, and other marine organisms are rich in bioactive compounds such as antioxidants, anti-inflammatory agents, and anti-cancer substances. The Sargassaceae family was chosen based on its unique properties and extensive research indicating its potential in cancer therapy. Sargassum species are rich in bioactive compounds, such as fucoidans, phlorotannins, and meroterpenoids, which have demonstrated significant anti-cancer activities in various studies. Fucoidans, for instance, have been shown to inhibit the proliferation of cancer cells by inducing cell cycle arrest and apoptosis, enhance the immune response against cancer cells by activating natural killer cells and macrophages, and inhibit cancer cell adhesion, invasion, and metastasis by modulating the expression of matrix metalloproteinases [37]. Phlorotannins are known for their strong antioxidant properties, which protect cells from oxidative stress, a factor in cancer progression. They also have anti-inflammatory effects, reducing inflammation linked to cancer development, and can induce apoptosis in cancer cells through various pathways, including the mitochondrial pathway [38]. Meroterpenoids exhibit cytotoxic effects specifically against cancer cells while sparing normal cells. They inhibit the formation of new blood vessels essential for tumor growth and metastasis and can enhance the efficacy of conventional chemotherapy drugs, potentially reducing the required dosage and associated side effects [39,40]. These unique properties and the extensive research supporting the anti-cancer potential of Sargassaceae phytochemicals justify their selection for this study. Integrating natural phytochemicals with conventional medicine offers a promising approach to enhance treatment outcomes and reduce adverse effects. Combining the cytotoxic effects of conventional therapies with the multi-targeted and synergistic actions of phytochemicals may provide a more comprehensive and effective approach to cancer management, particularly in metastatic disease.

Several studies have reported the potential therapeutic benefits of phytochemicals in the management of OSCC and BC. For instance, Wojtyłko et al. [41] investigated the anti-cancer properties of curcumin, a bioactive compound found in turmeric, and found that it inhibits the proliferation of cells and induces apoptosis in OSCC cells. Similarly, Cardona-Mendoza et al. [42] demonstrated the anti-metastatic effects of a polyphenol named resveratrol, naturally occurring in grapes and red wine, in breast cancer by suppressing cell migration and invasion. Furthermore, Jing et al. [43] reported the chemopreventive effects of epigallocatechin gallate (EGCG), a catechin found in extracts of green tea, against OSCC development through its antioxidant and anti-inflammatory properties. Additionally, Zhang et al. [44] investigated the potential of sulforaphane, a compound present in cruciferous vegetables, in inhibiting breast cancer metastasis by targeting epithelial-mesenchymal transition (EMT) signaling pathways. These findings underscore the promising role played by phytochemicals in the prevention and treatment of OSCC and BC.

To identify the metastatic protein target, we employed various computational approaches, including differential gene expression analysis, ontological assessment, and protein network construction. Based on data integration, 39 proteins (APH1A, BCL2L1, CA12, CTSD, FAAH, FDPS, HIF1A, HMGCR, HPRT1, HSD17B10, HSP90AA1, JAK3, KMT5A, MAPK1, MMP3, MSR1, PNP, PTGS1, PTPN1, SQLE, SYK, TDP1, TYMP, TYMS, VDR, VEGFA, CDC25C, KIF20B, MMP13, PDE4D, ALOX5AP, CDK5R1, MGAM, PTAFR, BLM, HPSE, IGF2R, INSR, and JAK1) were identified and assessed for their role in cancer metastasis. These 39 proteins were verified using the CMGene database, which indicated that 21 of them are involved in the process of metastasis. Then, through protein network analysis, HIF1A showed high-degree nodes and was designated as the desirable therapeutic target involved in the progression and metastasis. 

HIF1A (Hypoxia-inducible factor 1-alpha), a molecular marker plays a crucial role in embryonic vascularization and tumor angiogenesis, and it is often upregulated in human cancers through both hypoxic and non-hypoxic pathways and gene alteration as well [45]. 

In the current landscape of metastatic cancer research, hypoxia-inducible factor 1-alpha (HIF1A) has emerged as a critical player. HIF1A is a transcription factor that responds to low oxygen levels in the tumor microenvironment, a common characteristic of solid tumors. It regulates the expression of various genes involved in angiogenesis, metabolism, and survival, thereby promoting tumor growth and metastasis [46]. The overexpression of HIF1A has been linked to increased tumor aggressiveness and resistance to conventional therapies [47]. In OSCC, HIF1A contributes to the invasive and metastatic potential of cancer cells by modulating pathways that enhance cell motility and invasion [48]. Similarly, in breast cancer, particularly in triple-negative breast cancer (TNBC), HIF1A drives metastasis by regulating genes that promote epithelial-mesenchymal transition (EMT) and stemness [49]. Given its pivotal role in tumor progression and metastasis, HIF1A represents a promising therapeutic target. Targeting HIF1A could potentially inhibit the metastatic spread and improve the overall survival of patients with OSCC and breast cancer. Mirzaei et al. highlighted the significant role of HIF-1 in various aspects of breast cancer biology, including angiogenesis, metabolic reprogramming, stem cell maintenance, EMT (epithelial-mesenchymal transition), metastasis, invasion, radioresistance, and chemoresistance [50]. Ebright et al., found that brain metastases from breast cancer show higher HIF1A protein levels compared with primary breast tumors. Elevated hypoxic signaling in circulating tumor cells (CTCs) is linked to poorer overall survival in these patients, suggesting potential therapeutic targets [51]. Hegde et al., identified genes regulated by STAT3/HIF1A and EMT-specific transcription factors as novel predictors of metastasis in breast cancers [52]. Wang et al., reported that cell migration and invasion assays suggested that HIF1A-AS2 inhibition significantly depressed the migration and invasion of triple-negative breast cancer (TNBC) cells [53]. Yang et al. reported that HIF-1α promotes the proliferation and invasion of MCF-7 breast cancer cells by upregulating the expression of miR-210 [54]. Zhao et al. found that suppressing HIF-1α in 4T1 murine breast cancer cells led to a notable decrease in lung metastasis from the breast in mice [55]. This highlights HIF1A’s crucial role in promoting breast cancer metastasis, suggesting it is a potential target for therapeutic intervention. Therefore, we sought natural compounds that could potentially inhibit this therapeutic target.

Thus, phytochemicals from the sargassum species were collected from the CMNPD database and then docked against HIF1A. Among them, isonahocol D2 exhibited the highest binding affinity, with a G score of −7.1 kcal/mol. Also, it showed zero violations in ADME screening based on pharmacokinetics investigation. Moreover, the molecular dynamics simulation was well-supported based on the molecular docking of the HIF1A-isonahocol D2 complex, confirming the stable confirmation of the HIF1A target protein through isonahocol D2 inhibition. Particularly, the RMSD analysis showed fluctuations in the structure of the complex during its initial phase, with stabilization occurring after 50 ns. This suggests a dynamic behavior between various conformations. This stabilization implies that the complex eventually settles into an energetically favorable and structurally stable conformation with an average deviation of 3.6 Å. 

This stability is vital for the functional significance of the complex, ensuring the integrity of the binding and promoting effective interaction between HIF1A and isonahocol D2. Moreover, the RMSF analysis of the ligand identifies specific atoms within isonahocol D2 that are crucial for its interaction with HIF1A. This elucidates the molecular characteristics of the ligand that substantially contribute to its ability to bind to and regulate HIF1A activity. Furthermore, RMSF analysis highlighted distinct flexibility patterns among protein residues, particularly fluctuations at the active sites crucial for binding. Notably, residues at the N-terminal exhibited higher flexibility compared with the C-terminal, suggesting differential dynamics within the protein structure. The protein–ligand contact analysis revealed several types of interactions between HIF1A and isonahocol D2, including hydrogen bonds, hydrophobic bonds, and water bridges. The interactions between these hydrogen bonds play a crucial role in ligand binding, influencing drug specificity, metabolism, and adsorption. Hydrogen bonds involve the attraction between a hydrogen atom and an electronegative atom, which can be oxygen or nitrogen. In our analysis, amino acids LYS5, ASP13, ARG17, ARG24, ASN49, and SER50 formed hydrogen bonds with isonahocol D2. They contribute to the stability of complex protein–ligand molecules by burying hydrophobic regions away from the surrounding water molecules. In our analysis, the residues PHE21, ARG24, and ILE47 participate in hydrophobic bonding with isonahocol D2, forming interactions with its aromatic or aliphatic groups. Water bridges are hydrogen-bonded interactions mediated by a water molecule between the protein and the ligand. They provide additional stabilization to the protein–ligand complex. In our analysis, the residues LYS5, ASP14, THR20, LYS23, ARG24, PHE48, ASN49, and SER50 were involved in forming water bridges with isonahocol D2, facilitating indirect interactions through water molecules. Collectively, these results suggest the interactive stability of the HIF1A–isonahocol D2 complex at its binding site residues. Therefore, our findings propose isonahocol D2 as a potential therapeutic agent targeting HIF1A conditions of OSCC and breast cancer. Overall, this study proposes that isonahocol D2 would be the best drug candidate against the HIF1A therapeutic target for the metastasis of OSCC from breast cancer. 

This study identified HIF1A as a prominent metastasis target in OSCC metastasized from breast cancer, with isonahocol D2 showing strong binding affinity and favorable pharmacokinetic properties. To provide a more comprehensive view of potential treatment strategies, we also explored alternative therapeutic targets and compounds that could be considered alongside isonahocol D2. Cancer is a multifactorial disease characterized by complex interactions between various molecular pathways. Targeting a single pathway may not be sufficient to achieve optimal therapeutic outcomes due to the potential for resistance and the redundancy of signaling pathways. Therefore, a multifaceted approach that includes alternative therapeutic targets and compounds can enhance treatment efficacy and provide a more robust strategy for managing metastatic OSCC and breast cancer. For example, targeting HIF1A along with EGFR or VEGF inhibitors can disrupt both hypoxia adaptation and angiogenesis, which is crucial for tumor growth and metastasis [56]. For instance, combining isonahocol D2 with immune checkpoint inhibitors like PD-L1 inhibitors can enhance the immune system’s ability to recognize and destroy cancer cells, providing a more holistic approach to treatment [55]. Incorporating alternative therapeutic targets allows for the development of personalized treatment regimens based on the specific molecular profile of a patient’s tumor. This study presents several significant strengths: (1) an in-depth analysis of gene expression profiles in OSCC and BC, (2) the screening of natural compounds with potential anti-cancer activity, and (3) the identification of novel therapeutic targets that could drive future cancer treatments, particularly in addressing metastasis from breast cancer to OSCC. While these findings are promising, it is important to acknowledge the potential limitations of our study, such as the reliance on in silico methods and the need for experimental validation. The primary limitation of our study is the reliance on computational methods for target identification and validation. Although molecular docking and dynamics simulations provide valuable insights, experimental validation through in vitro and in vivo studies is essential to confirm the therapeutic potential of isonahocol D2. Future studies should focus on experimental validation to corroborate our computational findings.

## 4. Materials and Methods

### 4.1. Gene Expression Data Collection

We collected the RNA expression datasets related to OSCC and BC from Gene Expression Omnibus (https://www.ncbi.nlm.nih.gov/gds) (accessed on 6 June 2024) using the keywords like “OSCC”, “breast cancer”, “oral squamous cell carcinoma invasion”, and “breast cancer invasion”. We evaluated the collected datasets using the following inclusion and exclusion criteria. The following criteria for inclusion are (a) a dataset that consists of a minimum of three samples in a group; (b) datasets related to OSCC and BC; (c) a dataset containing gene expression profile assessed in Homo Sapiens; (d) a dataset with a relevant control group for comparative analysis. Similarly, the criteria for exclusion include the following: (a) samples without a control group; (b) DNA methylation or SNP arrays; and (c) studies other than OSCC and BC. From the collected datasets, we selected the GSE30784, GSE23558, GSE3524, and GSE10121 datasets for OSCC analysis. Likewise, the GSE10810, GSE15852, GSE20086, and GSE42568 datasets were selected for BC analysis. The Limma package (3.60.4) (R-program) was employed in performing differential analysis of gene expression with a log fold change of >1, with *p* < 0.05 [57]. Besides, genes were categorized as upregulated with *p*-value < 0.05 and logFC > 1 and as downregulated with *p*-value < 0.05 and logFC < −1. The Venny tool (2.1.0) (https://csbg.cnb.csic.es/BioinfoGP/venny.html) (accessed on 6 June 2024) was utilized to identify the commonly upregulated genes of OSCC and BC, which thus could serve as candidates to identify the metastatic signature.

### 4.2. Collecting and Screening Phytochemical Targets

The Comprehensive Marine Natural Products Database (CMNPD) is a valuable resource that comprises 31,561 unique components of natural marine origin along with over 13,000 marine organisms (https://www.cmnpd.org) (accessed on 6 June 2024) [58]. The compounds derived from sargassum species, a type of brown algae, were collected from the CMNPD. We identified a total of 147 unique compounds associated with sargassum species and downloaded their structural information in both Structure Data File (SDF) and Simplified Molecular Input Line Entry System (SMILES) formats. Next, we employed the Swiss Target Prediction (https://www.swisstargetprediction.ch/) (accessed on 6 June 2024), SuperPred (https://prediction.charite.de/) (accessed on 6 June 2024), and SEA Search (https://sea.bkslab.org/)(accessed on 6 June 2024) databases to predict the protein targets of the collected phytochemicals using their SMILES. Through the combined use of these databases, we predicted a total of 97 pharmacological targets for the 147 Sargassum-derived phytochemicals. These predicted targets were then mapped with the commonly upregulated genes identified from the OSCC and BC datasets. Such assessment provides a clue for possible sargassum-derived phytochemicals that could respond to the metastatic protein target.

### 4.3. Ontological Assessment and Network Construction

The functional significance of the interesting genes (OSSC, BC, and phytochemical targets) was assessed through the ShinyGo tool (2.1.0) (http://bioinformatics.sdstate.edu/go/) (accessed on 6 June 2024). ShinyGo is a bioinformatics resource that facilitates the interpretation of a given list of proteins through various functional annotations [59]. Specifically, the tool was employed to evaluate multiple aspects, including molecular function (MF), biological process (BP), and cellular component (CC). Subsequently, the interesting genes were assessed for their involvement in metastasis (signature) using the cancer metastasis gene database (CMGene, https://bioinfo-minzhao.org/cmgene/index.html) (accessed on 6 June 2024). Then, the interesting genes derived from OSCC, BC, and phytochemical targets were utilized to form a network of protein–protein interaction (PPI) through the STRING database of the Cytoscape software (3.9.1). Further, the cytoHubba plug-in of Cytoscape software analyzed the interaction network to calculate topological parameters such as the maximal clique centrality (MCC), the edge percolated component (EPC), the bottleneck, and the maximum neighborhood component (MNC), followed by the node connect degree [60]. However, we specifically focused on selecting the proteins based on their high degree of connectivity based on node-connect degree assessment [59]. Herein, degree refers to the number of direct connections between the proteins that form a core hub within a constructed network that enables the selection of a potential metastatic target.

### 4.4. Ligand and Target Protein Preparation

The preparation of ligands involved utilizing the LigPrep module within the Schrodinger suite (version 2013, Schrodinger. LLC, New York, NY, USA). The collected SDF structures of 147 phytochemicals sourced from sargassum were subjected to optimization by the LigPrep tool within Maestro v11.2 of the Schrödinger Suite, which was employed to prepare the structures of the ligands. This tool facilitates the optimization of molecular geometries and the generation of multiple conformers for each compound, accounting for different spatial orientations and conformational flexibility. To refine these structures and eliminate any undesirable interactions, we used the OPLS-3e force field for geometric optimization. We retrieved the three-dimensional (3D) structure of the “PD target” protein (PDB ID: 3KCX) from the Protein Data Bank (https://www.rcsb.org/) (accessed on 6 June 2024). This structure was then refined using the Protein Preparation Wizard in Maestro v11.2 from the Schrödinger Suite. The refinement involved several steps: assigning bond orders, forming bonds with metal ions, repairing missing side chains and disulfide bonds, adding hydrogen atoms, and removing extraneous water molecules and ligands. We calculated the protein’s pKa values with the PROPKA tool to optimize hydrogen bonding at pH 7.0. Finally, the protein structure underwent energy minimization using the OPLS-3e force field, which is designed for liquid simulations [61].

### 4.5. Molecular Docking

Molecular docking was executed using the Glide module in Maestro v11.2. Ligands with more than 500 atoms or 100 rotatable bonds were excluded from the analysis. A scaling factor of 0.8 was applied to van der Waals radii to accommodate ligands with partial charges greater than 0.15. The docking procedure was performed in Standard Precision (SP) mode, incorporating flexible and biased torsion sampling. Protonation states of ligands were adjusted with EPIK, and subsequent minimization was carried out to refine the docking results. Each ligand was assessed with a minimum of five distinct poses to ensure a detailed examination of binding interactions at the active site. Grid setup was carried out using the Glide Grid tool in Maestro, applying a scaling factor of 1 and a partial charge threshold of 0.25. A three-dimensional grid box was generated around the target protein’s active site residues using the Glide receptor grid panel. The identification of active sites was based on the available literature. The binding affinity of each phytochemical to the target was determined using the Glide scoring function [61].

### 4.6. ADME Screening

Pharmacokinetics plays a vital role in drug discovery by examining how drugs are absorbed, distributed, metabolized, and excreted, as well as their potential toxicity (ADMET). To evaluate these characteristics for selected natural compounds, the QikProp module within Maestro v11.2 from the Schrödinger Suite was employed. This tool provides predictions for several key pharmacokinetic and drug-likeness parameters:Lipinski’s Rule of Five: Assesses drug-likeness based on criteria such as molecular weight and lipophilicity;Caco-2 Cell Permeability (QPPCaco): Estimates the extent of intestinal absorption;Blood–Brain Barrier Penetration (QPlogBB): Predicts the compound’s ability to cross the blood–brain barrier and impact the central nervous system;Human Serum Albumin Binding (QPlogKhSa): Measures the extent of binding and distribution in the bloodstream;hERG Channel Blockade (QPlogHERG22): Evaluates the potential risk of cardiac toxicity related to the hERG potassium channel;Skin Permeability (QPlogKp): Estimates the compound’s ability to penetrate the skin.

The QikProp module assigns an ADMET-compliance star score to each compound, ranging from 0 to 5 that reflects the drug-likeness of the compound [62].

### 4.7. Molecular Dynamic Simulation Analysis

Following molecular docking, molecular dynamics simulations were carried out using the Desmond Molecular Dynamics System (Desmond-2019-4 module, Schrödinger, D. E. Shaw Research, New York, NY, USA, 2021) to assess the stability of the protein–ligand complex. The docked complex was solubilized in a cubic box filled with SPC water molecules, maintaining a minimum distance of 10 Å from the box edges. Sodium and chloride ions were added to neutralize the system, with long-range electrostatic interactions calculated using the particle-mesh Ewald method and van der Waals and Coulomb interactions truncated at 9.0 Å. Energy minimization was performed using the OPLS5 force field. The system was subjected to a 100-nanosecond simulation in the NPT ensemble, applying periodic boundary conditions and maintaining a temperature of 300 K and pressure of 1 atm using the Nosè–Hoover chain thermostat and Martyna–Tobias–Klein barostat [18,63]. Analysis of the simulation data included evaluating Root Mean Square Deviation (RMSD), Root Mean Square Fluctuation (RMSF), and protein–ligand interactions. Simulations were conducted in triplicate to ensure robustness, and the most stable and consistent outcomes were selected to ensure statistical validity and minimize random variability.

## 5. Conclusions

The study demonstrated that HIF1A plays a pivotal role in the metastatic advancement of OSCC originating from breast cancer. Targeting the HIF1A protein could provide effective therapeutic options to reduce the likelihood of breast cancer spreading to OSCC. The usage of isonahocol D2 might be advantageous for individuals with breast cancer who have a predisposition for acquiring oral squamous cell carcinoma via metastasis. To evaluate the effectiveness and safety, it is necessary to conduct preclinical validation and clinical studies on this molecular therapeutic target using isonahocol D2 in the future.

## Figures and Tables

**Figure 1 pharmaceuticals-17-01309-f001:**
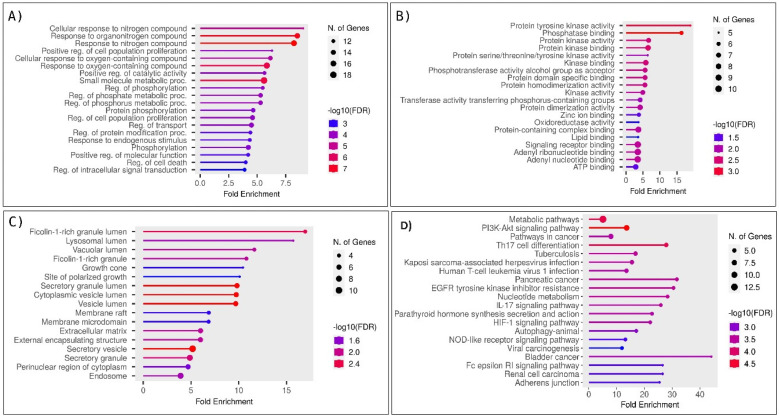
Enrichment and gene ontology analysis of 39 shared proteins involved in OSCC and breast cancer. (**A**) The biological process of 39 shared proteins identified their roles in cell proliferation, phosphorylation, and programmed cell death regulation. (**B**) Molecular function analysis revealed significant activities related to protein tyrosine kinase and transmembrane receptor protein tyrosine kinase activities. (**C**) Cellular component analysis indicated these proteins are integral to the plasma membrane. (**D**) Molecular pathway analysis showed their involvement in the PI3K-AKT, HIF-1 signaling, and metabolic pathways.

**Figure 2 pharmaceuticals-17-01309-f002:**
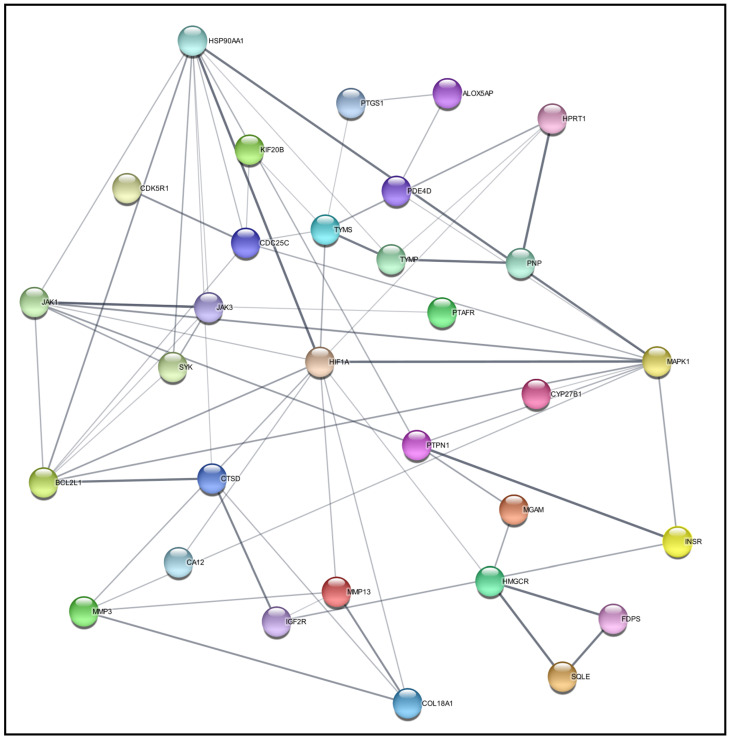
A protein–protein interaction (PPI) network of 39 targets for target screening.

**Figure 3 pharmaceuticals-17-01309-f003:**
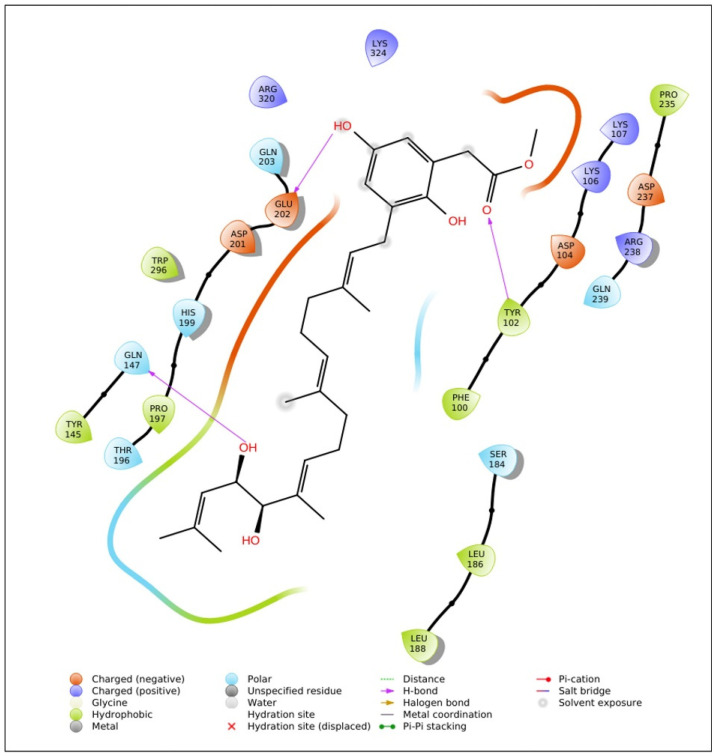
The molecular interaction plot of HIF1A and isonahocol D2 reveals potential binding modes with three hydrogen bonds formed at GLN147, GLU202, and TYR102.

**Figure 4 pharmaceuticals-17-01309-f004:**
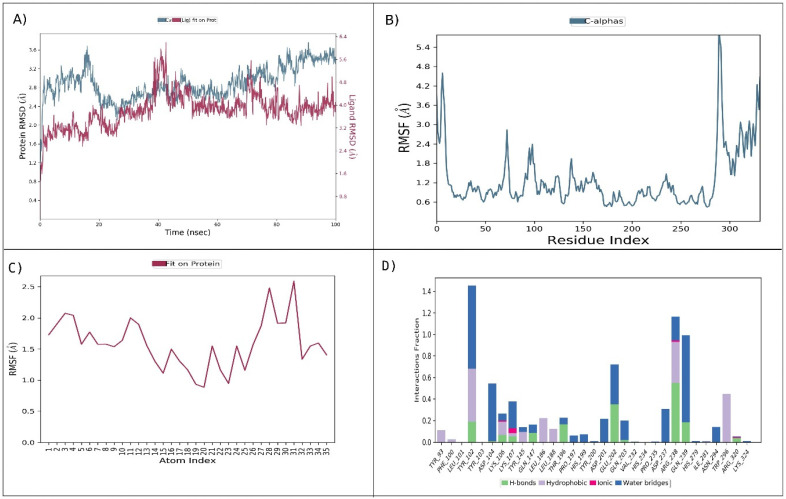
Molecular dynamics simulation analysis of HIF1A–isonahocol D2: (**A**) RMSD (C-alpha) analysis of HIF1A–isonahocol D2. (**B**) protein root-mean-square fluctuation (RMSF) analysis revealed residue fluctuations, with higher values at the N-terminal (5.4 Å) compared with the C-terminal (4.2 Å). (**C**) Ligand RMSF highlighted key atoms within isonahocol D2 contributing to interaction with HIF1A. (**D**) Protein–ligand contact map identified amino acids TYR102, ARG238, GLN239, GLU202, and ASP104 prominently involved in the interaction with isonahocol D2.

## Data Availability

Data will be made available on request.

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
