# Peer review of "A Network Pharmacology Identified Metastasis Target for Oral Squamous Cell Carcinoma Originating from Breast Cancer with a Potential Inhibitor from F. sargassaceae"

_pharmaceuticals, 2024, doi:10.3390/ph17101309_

Round 1

Reviewer 1 Report

Comments and Suggestions for Authors

1. It appears that all the authors share the same affiliation, so there is no need to repeat it.

2. The authors mentioned F. sargassaceae, it is not clear if they refer to 1 species or all species of to the alga family, please note that alga species should be written in italic.

3. Enhance the Introduction for Broader Context: While the introduction effectively outlines the study's purpose, it could benefit from a broader discussion on the current landscape of metastatic cancer research, particularly regarding OSCC and breast cancer. Including more context about the challenges in managing these cancers and why HIF1A is a particularly promising target would provide a stronger rationale for the study.

4. Clarify Methodological Details: Some sections, particularly the computational methods and the docking process, could be more detailed. Providing additional information on the selection criteria for the 39 proteins, the specific algorithms used in the molecular dynamics simulation, and the parameters for the ADME screening would enhance the reproducibility and transparency of the study.

5. While the discussion appropriately focuses on the study's findings, it should also acknowledge potential limitations, such as the reliance on in silico methods and the need for experimental validation. Additionally, discussing alternative therapeutic targets or compounds that could be considered alongside isonahocol D2 would provide a more comprehensive view.

6. The manuscript heavily relies on in silico methods without sufficient experimental validation, the authors used computational docking and molecular dynamics simulations, to identify and validate isonahocol D2 as a potential therapeutic agent. However, there is a lack of experimental validation, such as in vitro or in vivo studies, to support these findings. The absence of empirical data raises concerns about the real-world applicability and effectiveness of isonahocol D2 in targeting HIF1A and reducing metastasis.

7. The manuscript doesn't sufficiently discuss the study's limitations. In particular, relying heavily on computational methods without experimental validation is a significant drawback. It's important to include a more balanced discussion of these limitations to give a clearer picture of the study's findings and to help direct future research.

8. There is some ambiguity in methodological details and there is a lack of clarity and detail in the methodological sections, particularly concerning the selection process for the 39 proteins, the specific parameters used in molecular docking, and the criteria for ADME screening. This ambiguity can hinder the reproducibility of the study and raises concerns about the robustness of the findings.

9. The authors did not provide a detailed list of the specific phytochemicals tested from the Sargassaceae family, and they did not clearly explain the rationale for selecting this particular family of marine alga. This omission is a significant flaw in the manuscript for several reasons

a. Lack of Specificity in Phytochemical Selection: The manuscript mentions phytochemicals from the Sargassaceae family were screened against HIF1A, but it doesn't list the specific compounds or selection criteria. This omission makes it hard to assess the screening process's thoroughness and relevance.

b. Inadequate Justification for Selecting Sargassaceae: There is no strong rationale for choosing the Sargassaceae family over other sources of bioactive compounds. The selection seems arbitrary without discussing its unique properties or prior research that could justify this focus.

c. Missed Opportunity for a Broader Context: The authors missed the chance to compare Sargassaceae with other phytochemical sources known for anti-cancer properties. A comparative analysis would have enhanced the manuscript's scientific foundation and clarified the significance of their choice.

After careful consideration, I regret to inform you that your manuscript, in its current form, is unsuitable for publication.

Author Response

Comment 1: It appears that all the authors share the same affiliation, so there is no need to repeat it.

Response: We would like to clarify that the author affiliations have been drafted according to the journal’s guidelines, which request that we list the full affiliations of all contributing authors. This ensures clarity and proper acknowledgment of each author’s institutional association, even if they share the same affiliation. We appreciate your attention to detail and hope this explanation clarifies our approach.

Comment 2: The authors mentioned F. sargassaceae, it is not clear if they refer to 1 species or all species of to the alga family, please note that alga species should be written in italic.

Response: 

Thank you for the valuable comment. The term F. Sargassaceae is indeed a general reference to species within the genus Sargassum, where "F." denotes the family Fucales. Additionally, We have revised the manuscript and ensured that all scientific names are now italicized by taxonomic conventions.

Comment 3: Enhance the Introduction for Broader Context: While the introduction effectively outlines the study's purpose, it could benefit from a broader discussion on the current landscape of metastatic cancer research, particularly regarding OSCC and breast cancer. Including more context about the challenges in managing these cancers and why HIF1A is a particularly promising target would provide a stronger rationale for the study.

Response: As suggested, we have now incoporated all the suggested context and discussed elaborately in the Introduction about the purpose of the study and Discussion section on HIF1A as promising target.

Comment 4: Clarify Methodological Details: Some sections, particularly the computational methods and the docking process, could be more detailed. Providing additional information on the selection criteria for the 39 proteins, the specific algorithms used in the molecular dynamics simulation, and the parameters for the ADME screening would enhance the reproducibility and transparency of the study.

Response: The revised manuscript includes the methodology with comprehensive information. Additionally, the selection criteria for the 39 proteins are already discussed in detail in the Results section. Also, the algorithms used in the molecular dynamics simulation, and the parameters for the ADME screening was provided in the methodology section.

Comment 5: While the discussion appropriately focuses on the study's findings, it should also acknowledge potential limitations, such as the reliance on in silico methods and the need for experimental validation. Additionally, discussing alternative therapeutic targets or compounds that could be considered alongside isonahocol D2 would provide a more comprehensive view.

Response:  We have included the potential limitations of the study in the Discussion part with additional information on alternative therapeutic targets and compounds.

Comment 6: The manuscript heavily relies on in silico methods without sufficient experimental validation, the authors used computational docking and molecular dynamics simulations, to identify and validate isonahocol D2 as a potential therapeutic agent. However, there is a lack of experimental validation, such as in vitro or in vivo studies, to support these findings. The absence of empirical data raises concerns about the real-world applicability and effectiveness of isonahocol D2 in targeting HIF1A and reducing metastasis.

Response: Thank you for your feedback. We acknowledge the importance of experimental validation to support our in silico findings. Future research will include comprehensive in vitro and in vivo studies to confirm the therapeutic potential of isonahocol D2.

Comment 7: The manuscript doesn't sufficiently discuss the study's limitations. In particular, relying heavily on computational methods without experimental validation is a significant drawback. It's important to include a more balanced discussion of these limitations to give a clearer picture of the study's findings and to help direct future research.

Response: Thank you for your feedback. We have addressed the study’s limitations in the revised manuscript.

 Comment 8: There is some ambiguity in methodological details and there is a lack of clarity and detail in the methodological sections, particularly concerning the selection process for the 39 proteins, the specific parameters used in molecular docking, and the criteria for ADME screening. This ambiguity can hinder the reproducibility of the study and raises concerns about the robustness of the findings.

Response: The methodological section has been revised and detailed. Specifically, the selection process for the 39 proteins is discussed in detail in the Results section.

Comment 9: The authors did not provide a detailed list of the specific phytochemicals tested from the Sargassaceae family, and they did not clearly explain the rationale for selecting this particular family of marine alga. This omission is a significant flaw in the manuscript for several reasons.

  1. Lack of Specificity in Phytochemical Selection: The manuscript mentions phytochemicals from the Sargassaceae family were screened against HIF1A, but it doesn't list the specific compounds or selection criteria. This omission makes it hard to assess the screening process's thoroughness and relevance.

Response: Thank you for the comment. The specific phytochemicals screened in our study were sourced from the Comprehensive Marine Natural Products Database (CMNPD). We have now presented in a detailed list of these compounds in the supplementary table and the selection criteria in the revised manuscript.

  1. Inadequate Justification for Selecting Sargassaceae: There is no strong rationale for choosing the Sargassaceae family over other sources of bioactive compounds. The selection seems arbitrary without discussing its unique properties or prior research that could justify this focus.

Response:  Thank you for highlighting this point. We have now provided a comprehensive rationale for selecting the Sargassaceae family and information on their therapeutic properties in the revised manuscript (refer to discussion section).

  1. Missed Opportunity for a Broader Context: The authors missed the chance to compare Sargassaceae with other phytochemical sources known for anti-cancer properties. A comparative analysis would have enhanced the manuscript's scientific foundation and clarified the significance of their choice.

Response: Thank you for the comment. In the revised version, we have included a detailed comparative discussion of Sargassaceae with other known anti-cancer phytochemical sources.  

Comment 10: After careful consideration, I regret to inform you that your manuscript, in its current form, is unsuitable for publication.

Response: We have addressed all the comments and made comprehensive revisions to the manuscript. We believe these changes have significantly improved the quality of the manuscript, making it now suitable for publication.

Reviewer 2 Report

Comments and Suggestions for Authors

The article by Abdullah Alqarni et al. raises a very interesting topic. The authors approach the issue of oral squamous cell carcinoma that metastasizes from breast cancer in a modern way, using network pharmacology.

The introduction is a very good introduction to the subject. The goal is clearly defined.

I have no reservations about the methodology; the selection of methods to achieve the author's intended goal is correct.

I think a chart with a research scheme would be helpful and convenient for the readers. I suggest adding it.

The results are presented correctly. The number of figures is quite appropriate. However, Figures 1 and 4 are blurry. I suggest correcting them.
Similarly, in Figure 2, the names of the proteins are barely visible.

Discussion - I think it should be divided into sections. Writing in this way, in one continuous text, is difficult to get through.
The discussion should integrate more explicit references to figures or specific data points that support the conclusions. This would enhance clarity and allow readers to link the discussion to the presented results directly. I suggest rewriting the discussion a bit. The beginning of the discussion looks like an introduction. So, my main point concerns the organization of the discussion and its division into subsections.

It would be beneficial to state the study's limitations explicitly. This should be added. Additionally, providing more concrete suggestions for future research could add depth to the discussion.

Comments on the Quality of English Language

Minor editing of the English language is required.

Author Response

I think a chart with a research scheme would be helpful and convenient for the readers. I suggest adding it.

Comment 1: The results are presented correctly. The number of figures is quite appropriate. However, Figures 1 and 4 are blurry. I suggest correcting them.
Similarly, in Figure 2, the names of the proteins are barely visible.

Response: Sorry for the inconvenience. All the figures are now presented with better quality in the revised manuscript.

Comment 2: Discussion - I think it should be divided into sections. Writing in this way, in one continuous text, is difficult to get through.
The discussion should integrate more explicit references to figures or specific data points that support the conclusions. This would enhance clarity and allow readers to link the discussion to the presented results directly. I suggest rewriting the discussion a bit. The beginning of the discussion looks like an introduction. So, my main point concerns the organization of the discussion and its division into subsections.

Response: Thank you for the suggestion. The discussion has been reorganized into clear sections to enhance readability and includes explicit references to figures and data points to support the conclusions.

Comment 3: It would be beneficial to state the study's limitations explicitly. This should be added. Additionally, providing more concrete suggestions for future research could add depth to the discussion.

Response: Thank you for your suggestion. The limitations of the study have been stated in the revised manuscript. Additionally, we have included concrete suggestions for future research to provide greater depth and direction.

Reviewer 3 Report

Comments and Suggestions for Authors

Many References are missing

Results presentation will confuse the median Reader

Author Response

Comment 1: Many References are missing

Response: Thanks for the input. All references have been thoroughly checked and updated in the revised manuscript.

Comment 2: Results presentation will confuse the median Reader

Response: Thank you for comment. We have restructured the results section to improve clarity and readability, ensuring that the presentation is straightforward and accessible for all readers.

Comment 3: Comments on the Quality of English Language, Minor editing of the English language is required.

Response: Thanks for the input. We have reviewed and refined the manuscript to address the minor language issues and enhance overall clarity and readability.